# Linkages among Soil Properties and Litter Quality in Agroforestry Systems of Southeastern Brazil

**Priscila S. Matos** [1,*][ID], **Steven J. Fonte** [2][ID], **Sandra S. Lima** [1][ID], **Marcos G. Pereira** [1][ID], **Courtland Kelly** [2][ID], **Júnior M. Damian** [3][ID], **Marcelo A. Fontes** [4], **Guilherme M. Chaer** [4], **Felipe C. Brasil** [5] and **Everaldo Zonta** [1]

[1] Department of Soil Science, Federal Rural University of Rio de Janeiro, BR 465, Km 07, Seropédica, Rio de Janeiro 23897-000, Brazil; sandraslima01@gmail.com (S.S.L.); mgervasiopereira01@gmail.com (M.G.P.); ezonta@ufrrj.br (E.Z.)

[2] Department of Soil and Crop Sciences, Colorado State University, Fort Collins, CO 80523, USA; steven.fonte@colostate.edu (S.J.F.); courtland.kelly@colostate.edu (C.K.)

[3] Department of Soil Science, University of São Paulo, Av. Pádua Dias 11, Piracicaba, SP 13418-260, Brazil; damianjrm@usp.br

[4] National Agrobiology Research Center, BR-465, Km 7, Seropédica, Rio de Janeiro 23890-000, Brazil; marcelo.fontes@embrapa.br (M.A.F.); gchaer@cnpab.embrapa.br (G.M.C.)

[5] Ambiente Brasil, Av. Carlos Chagas Filho, 791, Ilha do Fundão, Rio de Janeiro 21941-904, Brazil; felipebrasil@ambientebrasil.net

* Correspondence: pri.eng.florestal@gmail.com; Tel.:+55-(21)-99595-5825

**Abstract:** Agroforestry systems have been promoted as a solution to address trade-offs between environmental conservation efforts and the need for increased agricultural productivity on smallholder farms in Brazil. However, the impact of land use change from degraded pasture to agroforestry on soil properties remains unclear. The objectives of this research were to: (1) assess soil chemical, physical and biological properties across distinct land uses (degraded pasture, agroforestry and secondary forest); and (2) understand relationships between litter quality, soil organic matter (SOM) and key soil quality parameters in the Brazilian Atlantic Rainforest. Soils, macroinvertebrates and litter were collected in April and September of 2018 under five land uses, including: three types of agroforestry systems, a degraded pasture and a secondary forest in Sapucaia, Rio de Janeiro, Brazil. Our results showed that soil properties clearly separated the three agroforestry systems plots (AS1, AS2, AS3) from the forest and pasture plots. Moreover, litter quality and SOM likely influence multiple biological and physiochemical soil properties under agroforestry systems and secondary forest. Our findings suggest that agroforestry systems can help support soil biological, chemical and physical properties and that the litter quality may be an important driver of their effects and potential contributions to soil restoration in the region.

**Keywords:** C:N ratio; enzymes; macroinvertebrates; pasture; restoration; secondary forest; soil degradation; soil organic matter; soil quality

## 1. Introduction

Degradation of agricultural lands around the globe threatens food security and the resilience of agricultural systems in the face of climate change [1]. In Brazil, data from the Ministry of the Environment indicate that there are about 52.3 million ha of degraded pasture [2], representing over half of the total pasture area in Brazil [3,4]. The main driver of soil degradation in Brazilian agricultural lands is water erosion, followed by acidification, compaction, salinization, pollution and desertification in semiarid areas [5]. Agroforestry systems have been recommended as a means to restore degraded

lands [6–8], with some studies suggesting that agroforestry systems containing native tree species can help facilitate secondary succession, similar to what happens in secondary forests [9–11].

In 2015, Brazil signed the Paris Agreement and committed to a 47% reduction in greenhouse gas emissions by 2030. A substantial portion of these climate change mitigation commitments relies on highly ambitious targets—restoring 15 million ha of degraded forests and 12 million ha of degraded pastures. Among the technologies suggested in this plan are agroforestry systems for recovery of degraded pasture [12]. In addition to greenhouse gas (GHG) reduction targets, recent environmental legislation (law number 12.651/12) requires rural landowners to maintain a portion of their lands (20% cover for areas within the Atlantic Forest Biome) with perennial vegetation cover (legal reserve). In this new legislation, agroforestry systems are recognized as a means to help farmers meet this requirement, while providing multiple socio-economic benefits.

Different types of agroforestry systems and practices lead to varying impacts on ecosystem services and soil quality. Simple agroforestry systems (with few tree species) may not meet restoration criteria as established by Brazilian law due to low levels of biodiversity and structural complexity that may not adequately provide the desired level of ecosystem services. Other more complex systems can be quite effective in supporting a range of ecological and economic functions [13]. In this regard, agroforestry systems with high biodiversity or 'successional' agroforests are often preferred over simpler farming systems. Some studies indicate positive effects of high species diversity and functional heterogeneity in agroforestry systems on soil chemical, physical and biological properties [14–16]; however, evaluation of different agroforestry systems remains scarce and merits further research.

More complex agroforestry systems exhibit notable similarities to natural forests due to their extensive tree cover and presence of a more developed litter layer [17]. Litter deposition in agroforestry systems is critical for maintenance soil organic matter (SOM) [18,19]. Leguminous species that are generally used in agroforestry systems contribute to natural regeneration because of their association with nitrogen (N)-fixing bacteria. Nitrogen-fixing plants increase the performance and fertility of agroforest soils by producing high quality leaf litter (i.e., low C:N ratio), which favors the release of N to the soil [20]. Practices associated with mixed agroforestry systems, such as the inclusion of vegetation that is structurally and taxonomically diverse, as well as continuous soil cover, are often associated with soil biological activity, including enhanced abundance and diversity of soil macrofauna [21,22]. Additionally, agroforestry systems can contribute to C sequestration in agricultural lands via storage of C in tree biomass and SOM [23–25].

Given the current widespread conversion of degraded pastures to agroforestry systems across Brazil, it is imperative to more fully understand how agroforestry systems may influence overall soil health and fertility [26]. This shift to agroforestry systems is likely associated with improved nutrient cycling and greater soil biological activity, with implications for multiple soil functions, but more research is needed. To address this knowledge gap, this study considered an experimental farm in southern Brazil, where diverse management practices had been implemented on plots with similar soil properties and management history prior to the establishment of three distinct agroforestry systems. In this context, we aimed to: (1) assess soil chemical, physical and biological properties across degraded pasture, different agroforestry systems and secondary forest, and (2) understand relationships between litter quality, SOM and key soil quality parameters.

## 2. Materials and Methods

### 2.1. Site Description and Land Uses

This study was conducted at the Arca de Noé Farm, an agroecological research station located near the city of Sapucaia, Rio de Janeiro, Brazil (21°59′42″ S, 42°54′52″ W; Figure 1). At roughly 800 m elevation, the site is characterized by dry winters and temperate summers (Cwb in the Köppen Climate Classification system), with mean monthly temperatures that vary between 17 °C and 32 °C (June and January; respectively) and a mean annual rainfall of 1451 mm. Soils at this site are predominantly

Ultisols [27] with a clay-loam texture. The region is largely comprised of massifs of highland hills and cliffs, with a natural vegetation generally dominated by the Atlantic Forest, which is characterized as Dense Ombrophylous Forest [28].

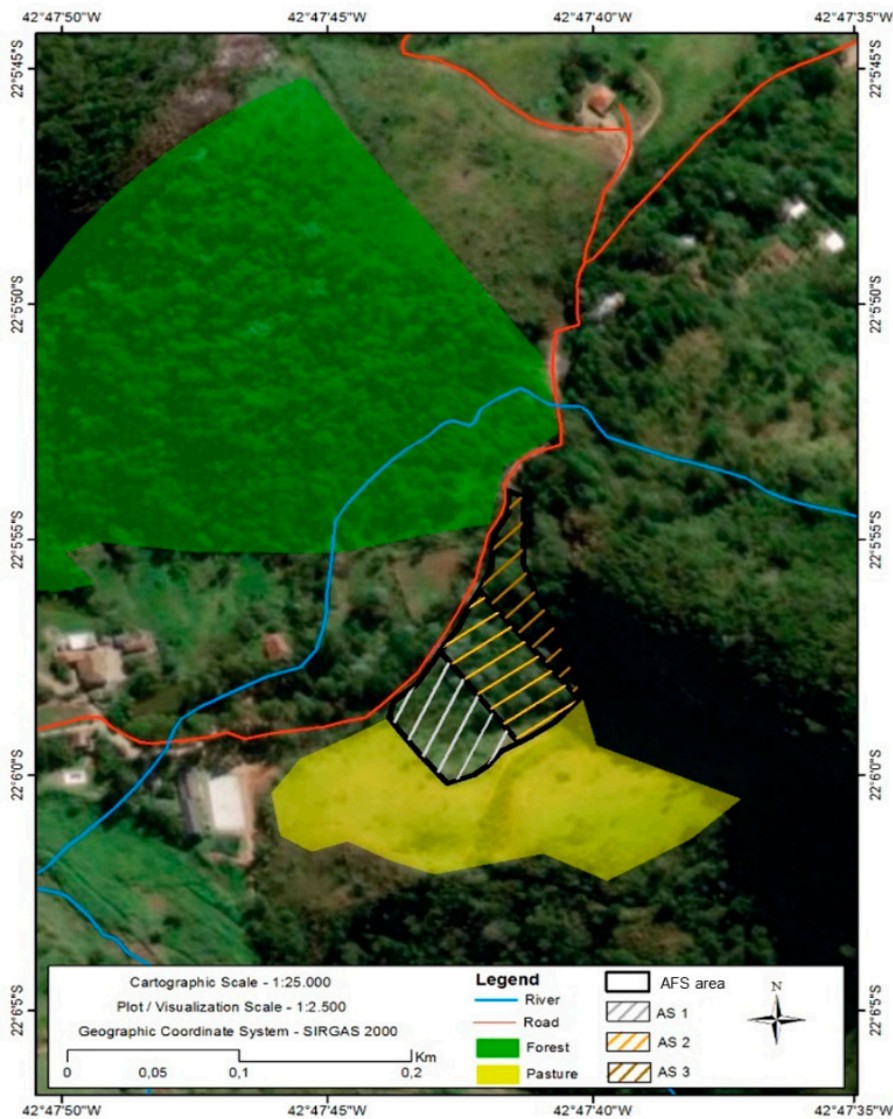

**Figure 1.** Arial view of the agroecological experimental station Arca de Noe Farm with the five studied land uses overlaid on top of the image. The farm is in the county of Sapucaia, Rio de Janeiro, Brazil.

In this study, we considered five existing land uses on the farm (Figure 1): (1) secondary forest (FOREST) dominated by semideciduous tree species (*Tibouchina granulosa*, *Lecythis pisonis* Cambess., *Centrolobium tomentosum* Guillem. ex Benth., *Inga* spp., *Schizolobium parahyba* (Vell.) Blake, *Peltophorum dubium*, *Hymenaea courbaril*, *Aspidosperma olivaceum* Müll. Arg., *Dalbergia nigra* (Vell.) Allemão ex Benth.); (2) pasture replanted with the grass *Urochloa decumbens*, under extensive grazing (PASTURE); (3) an agroforestry system characterized by the integration of banana and coffee with a mix of other fruit and timber species and other species to provide shade, biomass production and pollination services (AS1); (4) an agroforestry system focused on bananas and energy production (which also includes fruit trees and a mix of other trees and plants (AS2); and (5) a third agroforestry system focused on bananas and other fruits (AS3; see Table 1 for detailed species lists). Pasture was established by removal of native vegetation in 1995. In 2010, the agroforestry systems were planted on a portion of this existing pasture. These systems received a single application of rock phosphate

(fertilizer permitted in organic production) and cattle manure to the banana tree roots at the time of establishment. The secondary forest was included here as a reference and had an age of about 30 years since previous deforestation. All plots considered in this study were located on the same soil type and textural class and had similar slopes of roughly 30°.

**Table 1.** Plant species present in each of the three agroforestry systems established in 2010 at the Arca de Noé farm, Sapucaia, Rio de Janeiro, Brazil.

| System | Scientific Name | Family | Common Name | Function | Introduction |
|---|---|---|---|---|---|
| AS1 | *Musa paradisiaca* L. | *Musaceae* | banana | fruit production | planted |
| | *Coffea arabica* | *Rubiaceae* | coffee | grain production | planted |
| | *Carica papaya* L. | *Caricaceae* | papaya | fruit production | planted |
| | *Morus nigra* L. | *Moraceae* | black mulberry | fruit production | planted |
| | *Eugenia uniflora* L. | *Myrtaceae* | Brazilian cherry | fruit production | planted |
| | *Inga edulis* Mart. | *Fabaceae* | ice-cream bean | fruit production | planted |
| | *Myrciaria glazioviana* (Kiaersk.) G. M. Barroso ex Sobral | *Myrtaceae* | "cabeludinha" * | fruit production | planted |
| | *Malpighia glabra* L. | *Malpighiaceae* | "acerola" * | fruit production | planted |
| | *Mangifera indica* L. | *Anacardiaceae* | mango | fruit production | planted |
| | *Psidium guajava* L. | *Myrtaceae* | guava | fruit production | natural/regenerated |
| | *Hymenaea courbaril* L. | *Fabaceae* | "jatobá" * | timber production | planted |
| | *Tithonia diversifolia* (Hemsl.) A. Gray | *Arecaceae* | mexican-sunflower | biomass production | planted |
| | *Solanum mauritianum* Scop | *Solanaceae* | "fumo-bravo" * | shade and biomass production | natural/regenerated |
| | *Trema micrantha* (L.) Blume | *Cannabaceae* | "trema" * | shade and biomass production | natural/regenerated |
| | *Sapium glandulatum* (Vell.) Pax | *Euphorbiaceae* | "burra-leiteira" * | shade and biomass production | natural/regenerated |
| | *Vernonia polycephala* Less. | *Asteraceae* | "assa-peixe" * | pollination services | natural/regenerated |
| AS2 | *Musa acuminata* | *Musaceae* | banana | fruit production | planted |
| | *Musa paradisiaca* L. | *Musaceae* | banana | fruit production | planted |
| | *Jatropha curcas* L. | *Euphorbiaceae* | physic nut | energy production | planted |
| | *Persea Americana* Mill. | *Lauraceae* | avocado | fruit production | planted |
| | *Morus nigra* L. | *Moraceae* | black mulberry | fruit and biomass production | planted |
| | *Inga edulis* | *Fabaceae* | ice-cream bean | fruit and biomass production | planted |
| | *Eriobotrya japonica* (Thunb.) Lindl. | *Rosaceae* | "nêspera" * | fruit production | planted |
| | *Carica papaya* L. | *Caricaceae* | papaya | fruit production | planted |
| | *Mangifera indica* L. | *Anacardiaceae* | mango | fruit production | planted |
| | *Eugenia uniflora* | *Myrtaceae* | cherry | fruit production | planted |
| | *Tithonia diversifolia* (Hemsl.) A. Gray | *Arecaceae* | Mexican-sunflower | biomass production | planted |
| | *Solanum mauritianum* Scop | *Solanaceae* | "fumo-bravo" * | shade and biomass production | natural/regenerated * |
| | *Piper aduncum* L. | *Piperaceae* | "aperta-ruão" * | biomass production | natural/regenerated |
| | *Vernonia polycephala* Less. | *Asteraceae* | "assa-peixe" * | pollination services | natural/regenerated |

**Table 1.** *Cont.*

| System | Scientific Name | Family | Common Name | Function | Introduction |
|---|---|---|---|---|---|
| | *Musa acuminata* | *Musaceae* | banana | fruit production | planted |
| | *Musa paradisiaca* L. | *Musaceae* | banana | fruit production | planted |
| | *Mangifera indica* L. | *Anacardiaceae* | mango | fruit production | planted |
| | *Artocarpus heterophyllus* | *Moraceae* | jack fruit | fruit production | planted |
| | *Citrus* sp. | *Rutaceae* | cravo lemon | fruit production | planted |
| AS3 | *Plinia trunciflora* (O. Berg) Kausel | *Myrtaceae* | jabuticaba | fruit production | planted |
| | *Campomanesia phaea* (O. Berg.) Landrum | *Myrtaceae* | cambuci | fruit production | planted |
| | *Solanum mauritianum* Scop | *Solanaceae* | "fumo-bravo" * | shade and biomass production | natural/regenerated |
| | *Piper aduncum* L. | *Piperaceae* | "aperta-ruão" * | biomass production | natural/regenerated |
| | *Tithonia diversifolia* (Hemsl.) A. Gray | *Arecaceae* | mexican-sunflower | biomass production | planted |

* natural/regenerated: species that were already in the area when the system was implemented. Species identified according with Flora do Brasil (2018).

## 2.2. Soil, Litter and Arthropod Sampling

Sampling was conducted in 2018 at 2 separate time points, in the rainy season (April) and dry season (September), to assess a suite of soil biological, chemical and physical properties within each land use (e.g., forest, pasture and agroforestry systems). A single transect was laid out in each of the land use types, and 4 sampling plots (6 × 8 m) were established approximately 15 m apart along the transect. The 4 plots within each land use were considered replications. While we recognize that the lack of true replication limits interpretation of our findings, we note that similar sampling designs have been used in previous studies evaluating the effect of land use on soil properties [29–32] and that this approach is important for studying unique management systems where replicated, randomized field trials are not feasible. Within each sampling plot, 4 subsamples of soil (0–10 cm depth) were collected using a shovel (~5 m spacing between subsamples) and combined to generate 1 composite sample per sampling plot per season. A portion of each composite sample was kept cool for transport to the laboratory at the Federal Rural University of Rio de Janeiro (Seropédica, Brazil), where it was stored at 4 °C (for <2 weeks) until analysis of microbiological parameters. The rest of each composite sample was air-dried, sieved to 2 mm, and analyzed for chemical properties.

Physical parameters were evaluated only in the rainy season (April). Bulk density (BD) was measured for the 4 subsamples per sampling point by inserting a metal cylinder ring (5 cm diameter) vertically into the soil to a 10 cm depth. Soil from within each ring was returned to the lab for separation into soil, rocks and large roots, and then dried at 105 °C, and weighed [33]. For evaluation of water-stable aggregation, 4 soil cores (10 cm diameter) were collected to a depth of 10 cm in each sampling plot and combined into 1 composite sample. Field-moist soil was passed through an 8 mm sieve by gently breaking soil clods along natural planes of fracture, and then air-dried for subsequent analyses.

The litter and soil dwelling arthropods were evaluated at both sampling times using pitfall traps adapted from Moldenke [34]. The traps consisted of plastic containers (10 cm diameter) that were inserted 10 cm deep into the soil such that they were level with the surface level. In total, 2 replicate traps were installed in each plot, for a total of 8 traps per land use system per sampling time. For both sampling times, traps remained in the field during a 9-day period and specimens collected in each trap were returned to the lab, stored in 70% ethanol and identified to the level of order, class or family, based on Gallo [35] and Dindal [36]. Abundance for each taxon was averaged across two traps in each

sampling plot and reported as individuals trap-1 day-1. Diversity was evaluated using both species richness (S, number of taxonomic groups) and the Shannon Index (H) [37] for each plot considering the number of unique taxonomic groups encountered by the 2 traps per plot, that is, average number of species per sampling plot in each land use system. A total of 40 traps were used to assess litter and soil dwelling arthropods in all land uses at each sampling time. The litter biomass was quantified by collecting the organic material at the soil surface that had not decomposed following the procedures proposed by Sanqueta [38]. A wooden square with internal area of 0.25 m$^2$ was placed ~1 m away from the pitfall traps within each sampling plot, for a total of 8 samples per land use system at each sampling time. This material was dried at 65 °C to constant weight, weighed and crushed in a mill for further chemical analysis.

### 2.3. Soil Microbial Measurements

For analysis of microbial biomass, Microbial Biomass Carbon (MBC) and Microbial Biomass Nitrogen (MBN), refrigerated soil (stored at 4 °C) was passed through a 2 mm sieve and 2 subsamples (20 g each) were weighed for each sampling point. One of these subsamples was fumigated with chloroform by 24 h and then shaken for 30 min with $K_2SO_4$ (0.5 mol L$^{-1}$), while the other was not fumigated and submitted to the same extraction procedure [39,40]. The estimation of C in microbial biomass was conducted with colorimetric determination [41]. Quantification of N in microbial biomass was performed according to the methods of Brookes et al. [42] by steam distillation (Kjeldahl), followed by acid-base volumetry with sulfuric acid as a titrator.

Soil basal respiration (Sbresp) for each sampling plot was assessed using soil respiration on duplicate 50 g subsamples of refrigerated soil [43]. Samples were stored in 100 mL flasks and incubated in glass jars with a volume of 3 L together with 10 mL of 1 mol L$^{-1}$ NaOH solution for 143 h (April) and 162 h (September). After the incubation period, the $CO_2$ trapped by the NaOH solution was precipitated with 2 mL of barium chloride (10%) in water and titrated with HCL (0.5 mol L$^{-1}$), using phenolphthalein (1%) as an indicator in alcoholic medium. The values of accumulated $CO_2$ were expressed in μg of C per g of dry soil.

Enzyme activity was assessed via quantification of β-glucosidase (C-cycle) and acid phosphatase (P-cycle). In addition, the total enzyme activity was evaluated by analyzing the hydrolysis of fluorescein diacetate (FDA). β-glucosidase activity was analyzed according to [44], using 1.0 g of fresh soil and the substrate p-nitrophenyl-β-D-glucoside (0.05 mol L$^{-1}$). Analysis of acid phosphatase activity was conducted with 1.0 g fresh soil, using p-nitrophenyl-sulfate as a substrate (0.05 moI L$^{-1}$) [44]. Colorimetric determination of was conducted in a spectrophotometer at 410 nm. Results were expressed in μmol·g$^{-1}$ h$^{-1}$ p-nitrophenyl. Analysis of fluorescein diacetate (FDA) hydrolysis was conducted according to Schnürer and Rosswall [45] and modified by Dick et al. [46], using 1.0 g soil fresh and FDA stock solution. Samples were read in a spectrophotometer at 490 nm to determine the amount of hydrolyzed fluorescein and results expressed in μg fluorescein g$^{-1}$ soil h$^{-1}$.

### 2.4. Soil Physicochemical Analyses

Soil organic carbon (SOC) was quantified by the oxidation of organic matter using a solution of potassium dichromate in acid medium, with an external source of heat [47]. Available P and K were evaluated using a Mehlich$^{-1}$ extractant ($H_2SO_4$ 0.0125 mol L$^{-1}$ + HCl 0.05 mol L$^{-1}$), while exchangeable $Ca^{2+}$, $Mg^{2+}$ and $Al^{3+}$ were extracted with KCl (1 mol L−1). The concentrations of these elements in the soil samples were determined by titration (P, $Al^{3+}$, K, $Ca^{2+}$ and $Mg^{2+}$). Cation exchange capacity (CEC; cmolc kg$^{-1}$) and pH was analyzed in a 1:5 suspension of soil and deionized water [33]. Analysis of permanganate oxidizable carbon (POXC) was conducted on 2.5 g air-dried soil based on the method of [48].

Water-stable aggregation (WSA) was determined using a Yoder wet-sieving apparatus [49]. For the evaluation of the aggregate distribution, 25 g of the air-dried, 8-mm sieved soil was transferred to the top of a set of sieves with 2.00, 1.00, 0.50, 0.25 and 0.105 mm mesh sizes, moistened with spray and

subjected to vertical oscillation in the Yoder apparatus, for 15 min. The material retained on each sieve was then rinsed into separate Petri dishes and dried in an oven at 65 °C. Mean weight diameter (MWD) of the aggregates was calculated according to van Bavel [50] by summing the proportions of soil in each size class multiplied by the corresponding average size of aggregates in each class.

Soil texture was determined by the pipette method [33]. In the first step, chemical dispersion was performed using NaOH 0.1 mol L$^{-1}$ as a dispersing agent, following the methodology used by Ruiz [51]. The second step consisted of mechanical dispersion by shaking at 60 rpm for a period of 16 h. Total clay (diameter <0.002 mm) and sand (diameter 2 to 0.05 mm) contents were obtained, respectively, by pipetting and sieving, while the silt content (diameter between 0.05 to 0.002 mm) was calculated by the difference.

### 2.5. Litter Nutrient Analysis

Litter samples were ground and evaluated for total C and N by dry combustion of 5.0 ± 0.1 mg samples using an elemental analyzer Perkin Elmer 2400 CHNS. Litter macro- and micronutrient concentrations were evaluated via the digestion method USEPA 3051A [52], which was conducted in a closed system using microwave radiation in a MARS Xpress® device. All analyses were performed in triplicate and used high purity acids (P.A.) and Milli-Q water for dilution. The concentrations of P, K, Mg and Ca in the extracts resulting from digestion were determined P by colorimetry, K by flame photometry, Ca and Mg by atomic absorption spectrometry using a Varian SpectrAA 55B device.

### 2.6. Statistical Analyses

One-way ANOVA with Tukey tests were used to compare soil properties and macrofauna communities among the 5 land uses separately for the rainy season (April) and dry season (September) sampling times. Data were ln transformed as needed to meet the assumptions of homoscedasticity and normality. All univariate tests were carried out using R statistical software [53]. Given that the 5 land uses were not replicated across multiple treatment plots, we considered the 4 sampling plots within each management type as replicates to explore differences between land use plots. While we understand that this experimental design limits broad causal inferences about management, valuable insight can be gained by exploring differences between the land uses and relationships between the variables measured.

For each data set (soil chemical, physical properties and microbiological properties), principle component analysis (PCA) together with between-class PCA were used to explore relationships between variables within a data set and multivariate differences between land uses at each time point. Highly collinear variables were omitted from the PCA and associated land use comparisons. Ordination and visualization of soil fauna communities was conducted using nonmetric multidimensional scaling (NMDS). Bray–Curtis distances were calculated between samples using the dominant soil taxa. Nonmetric multidimensional scaling (NMDS) ordinations were plotted for the distance matrices, and correlations between environmental variables and NMDS axes were calculated and included as arrows in plots if significant ($p < 0.05$). Treatment effects were tested using the ADONIS method of permutational multivariate analysis with 999 permutations. All multivariate analyses were completed in R using the vegan package [54] and ade4 library within the R environment [53,55].

In order to understand the role of SOM and litter quality in driving multiple soil quality parameters, we used multiple linear regression with each soil variable as the response, and the following model terms: SOM, sampling time and the sampling time by SOM interaction. Analysis was repeated with litter C:N ratio replacing SOM. With these models we were able to understand the relationships between SOM, litter C:N and multiple soil biological, chemical and physical parameters, while accounting for differences between the April and September evaluations. Data were ln transformed as needed to meet model assumptions and these analyses were conducted in JMP 14.0 statistical software [56].

## 3. Results

### 3.1. Soil Chemical and Physical Properties

The land use plots differed significantly in soil physical and chemical properties for both seasons evaluated. In the rainy season, the agroforestry systems generally contained higher levels of SOM, available P, Ca, Mg, K, CEC, pH than soils under forest or pasture management. In the dry season the same tendencies were apparent, but differences were only significant for available P, Ca and Mg (Table 2). Additionally, pH tended to be lower and $Al^{3+}$ levels higher in pasture and forest. The pasture system also presented higher bulk density than the agroforestry systems and secondary forest. Aggregate stability (MWD) in the rainy season was generally higher in the forest compared to the other land uses, but only significantly higher than AS3 (Table 2).

Similar to the findings from ANOVA, multivariate differences, using between-class PCA, of soil chemical and physical properties clearly separated soils in the three agroforestry systems plots (AS1, AS2, AS3) from the forest and pasture plots in rainy season ($p = 0.001$ by Monte Carlo Permutation test; 56.32% of randtest observation; Figure 2A). In the rainy season, the first and second principle component (PC) axes explained 39.72% and 20.87% of the variability, respectively. The agroforestry systems plots were separated from the other land uses mainly along PC1 and were associated with higher values of pH, P and SOC, while forest was associated with higher MWD, and pasture was associated with higher values of BD (Figure 2A). In the dry season PC 1 was associated with available P, pH and soil texture and explained 43.67% of the variability, while PC 2 was associated with SOC and CEC explained 20.66% of the variability. The agroforestry systems were associated higher values of pH, P and clay and mainly separated from the forest and pasture plots along PC 1 ($p = 0.002$ by Monte Carlo Permutation test; 43.47% of randtest observation; Figure 2B).

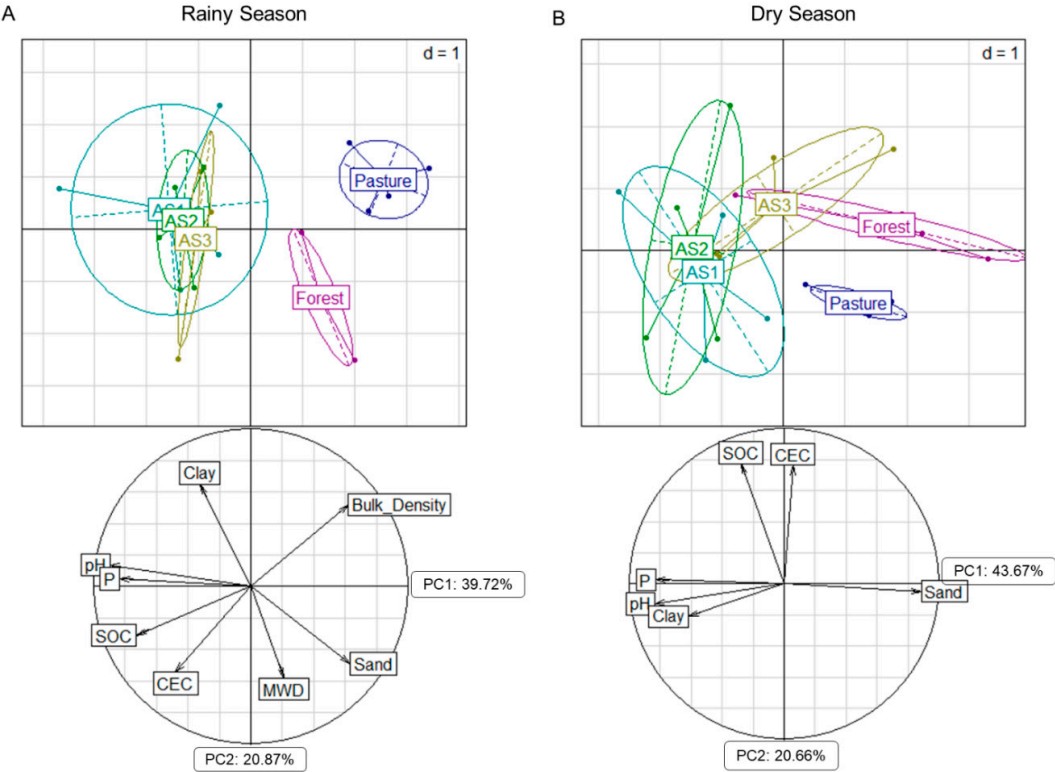

**Figure 2.** Between-class analysis of the 5 different land use using soil chemical and physical properties in (**A**) rainy season and (**B**) dry season ($p = 0.001$ and $p = 0.002$, for group separation in each time period, respectively, by Monte Carlo permutation test). Variable-correlation circle of soil chemical and physical properties in (**A**) rainy season and (**B**) dry season. See Table 2 for additional explanation of abbreviations.

**Table 2.** Mean values for soil chemical and physical properties sampled on an experimental farm in the county of Sapucaia, Rio de Janeiro, Brazil at two time points, in the rainy season (April) and dry season (September) of 2018.

| Soil Variable | Rainy Season (April) | | | | | | Dry Season (September) | | | | | |
|---|---|---|---|---|---|---|---|---|---|---|---|---|
| | Forest | Pasture | AS1 | AS2 | AS3 | *p*-Value | Forest | Pasture | AS1 | AS2 | AS3 | *p*-Value |
| pH | 4.5 [c] | 4.6 [c] | 5.7 [a] | 5.3 [ab] | 5.2 [b] | <0.001 | 4.5 [a] | 4.8 [a] | 5.1 [a] | 5.2 [a] | 4.8 [a] | |
| | *0.1* | *0.1* | *0.3* | *0.2* | *0.1* | | *0.4* | *0.1* | *0.2* | *0.4* | *0.4* | |
| SOC (g kg$^{-1}$) | 23.5 [ab] | 23.1 [b] | 28.0 [a] | 25.3 [ab] | 26.4 [ab] | 0.039 | 21.6 [a] | 21.4 [a] | 24.2 [a] | 22.6 [a] | 24.5 [a] | |
| | *0.44* | *1.9* | *2.26* | *2.12* | *3.4* | | *0.858* | *1.14* | *4.46* | *4.45* | *1.13* | |
| POXC (mg kg$^{-1}$) | 653 [a] | 653 [a] | 788 [a] | 1092 [a] | 826 [a] | | 786 [ab] | 576 [b] | 1121 [a] | 747 [ab] | 660 [b] | 0.023 |
| | *333* | *371* | *126* | *112* | *197* | | *234* | *101* | *38.7* | *225* | *323* | |
| Avail.P (mg kg$^{-1}$) | 26.5 [ab] | 22.5 [b] | 29.5 [a] | 31.5 [a] | 28.5 [a] | 0.002 | 20.4 [bc] | 18.8 [c] | 25.6 [a] | 27.7 [a] | 24.2 [ab] | <0.001 |
| | *0.6* | *0.6* | *4.7* | *3.1* | *1* | | *2.4* | *1.03* | *2.3* | *1.6* | *1.1* | |
| Ca$^{2+}$(meq 100 mg$^{-1}$) | 0.8 [b] | 0.6 [b] | 3 [a] | 2.9 [a] | 3 [a] | <0.001 | 0.8 [c] | 0.7 [c] | 2.4 [b] | 4.4 [a] | 3.4 [ab] | <0.001 |
| | *0.2* | *0.1* | *0.4* | *0.7* | *0.4* | | *0.3* | *0.2* | *0.3* | *0.9* | *0.6* | |
| Mg$^{2+}$(meq 100 mg$^{-1}$) | 0.7 [b] | 0.5 [b] | 2.0 [a] | 2.0 [a] | 2.1 [a] | <0.001 | 0.7 [b] | 0.4 [b] | 2.2 [a] | 1.7 [a] | 1.4 [a] | <0.001 |
| | *0.2* | *0.1* | *0.2* | *0.2* | *0.3* | | *0.3* | *0.1* | *0.1* | *0.5* | *0.1* | |
| K$^{+}$(meq 100 mg$^{-1}$) | 0.2 [d] | 0.1 [d] | 0.6 [b] | 0.5 [c] | 0.7 [a] | <0.001 | 0.2 [c] | 0.1 [c] | 0.6 [a] | 0.5 [a] | 0.4 [b] | <0.001 |
| | *0.08* | *0.01* | *0.08* | *0.03* | *0.04* | | *0.1* | *0.02* | *0.1* | *0.1* | *0.04* | |
| Na$^{+}$(meq 100 mg$^{-1}$) | 0.1 [a] | 0.1 [a] | 0.1 [a] | 0.1 [a] | 0.1 [a] | | 0.03 [b] | 0.04 [a] | 0.04 [a] | 0.04 [a] | 0.03 [ab] | <0.001 |
| | *0* | *0.01* | *0.01* | *0* | *0.01* | | *0.01* | *0.04* | *0.01* | *0* | *0.01* | |
| Al$^{3+}$(meq 100 mg$^{-1}$) | 1.5 [a] | 1.6 [a] | 0.3 [b] | 0.3 [b] | 0.2 [b] | <0.001 | 1.7 [a] | 1.6 [a] | 0.3 [c] | 0.4 [b] | 0.4 [bc] | <0.001 |
| | *0.2* | *0.06* | *0.2* | *0.2* | *0.2* | | *0.4* | *0.1* | *0.1* | *0.1* | *0.1* | |
| CEC (cmol kg$^{-1}$) | 16.9 [ab] | 15.8 [b] | 16.7 [ab] | 18.2 [ab] | 19.0 [a] | 0.019 | 16.5 [a] | 14.2 [a] | 13.8 [a] | 16.1 [a] | 16.1 [a] | |
| | *1* | *0.9* | *2.1* | *1.2* | *0.8* | | *1.5* | *0.5* | *1.4* | *1.4* | *1.4* | |
| BD (g m$^{-3}$) | 1.5 [b] | 1.8 [a] | 1.6 [b] | 1.5 [b] | 1.5 [b] | <0.001 | | | | | | |
| | *0.04* | *0.04* | *0.1* | *0.1* | *0.04* | | | | | | | |
| Clay (%) | 28.1 [a] | 30.3 [a] | 30.5 [a] | 33.8 [a] | 31.4 [a] | | | | | | | |
| | *3.3* | *3.3* | *1.1* | *5.1* | *4.7* | | | | | | | |
| Sand (%) | 55.8 [a] | 55.7 [a] | 40.01 [a] | 46.9 [a] | 49.6 [a] | | | | | | | |
| | *14.1* | *5.9* | *3.8* | *5* | *8.2* | | | | | | | |
| MWD (mm) | 4.7 [a] | 4.5 [ab] | 4.5 [ab] | 4.5 [ab] | 4.3 [b] | 0.025 | | | | | | |
| | *0.04* | *0.1* | *0.3* | *0.02* | *0.1* | | | | | | | |

Values in italics below each mean represent the standard error from four measurements in each plot. Means with different letters have significantly different values according to Tukey tests. Abbreviations: Avail., available; SOC, soil organic carbon; POXC, permanganate oxidizable carbon, CEC, cation exchange capacity; BD, bulk density; MWD, mean weight diameter; AS1, agroforestry system 1; AS2, agroforestry system 2; AS3, agroforestry system 3.

### 3.2. Microbiological Properties

The land uses differed significantly in soil microbiological properties in the two periods evaluated (Table 3). In the rainy season, agroforestry systems and pasture showed higher levels of MBC than the forest. The highest values of MBN were observed for AS3, pasture, forest and lowest values in AS1 and AS2. Regarding the activity of enzymes, β-glucosidase had higher values in agroforestry and forest plots than in the pasture plot and acid phosphatase was higher in the forest than in other land uses. Total enzymatic activity (FDA) was higher in pasture and forest than in agroforestry systems in the rainy season. In the dry season, agroforestry systems indicated an increase in microbial biomass, and greater microbial activity (Sbresp) and FDA than in forest and pasture. Additionally, phosphatase tended to be higher under forest than agroforestry systems and pasture. The other variables evaluated showed no difference between treatments in the dry season (Table 3).

Between-class PCA analysis of soil microbiological properties reinforced univariate ANOVA findings, clearly separating the agroforestry systems plots from the forest and pasture in the rainy season (*p* = 0.001 by Monte Carlo Permutation test; 72.43% of randtest observation; Figure 3A) and in the dry season (*p* = 0.001 by Monte Carlo Permutation test; 59.35% of randtest observation; Figure 3B). In the rainy season there was a clear separation of all management plots (randtest simulated *p*-value = 0.001). PC1 explained 41.49% of the variability, was associated with microbial biomass N (MBN), β-glucosidase activity, microbial activity (Sbresp) and total enzymatic activity (FDA), and separated the agroforestry system plots from the pasture plots. PC2 (explaining 30.24% of the variability) was strongly associated with phosphatase activity and microbial biomass C (MBC) and separated all the management plots from the forest (Figure 3A). In the dry season, PC1 and PC2 explained 48.66% and 18.24% of the variability, respectively (Figure 3B). Again, PC1 was associated with β-glucosidase activity, Sbresp, and FDA, with the addition of MBC for this time point, and separated the agroforestry systems plots from the forest and pasture (Figure 3B).

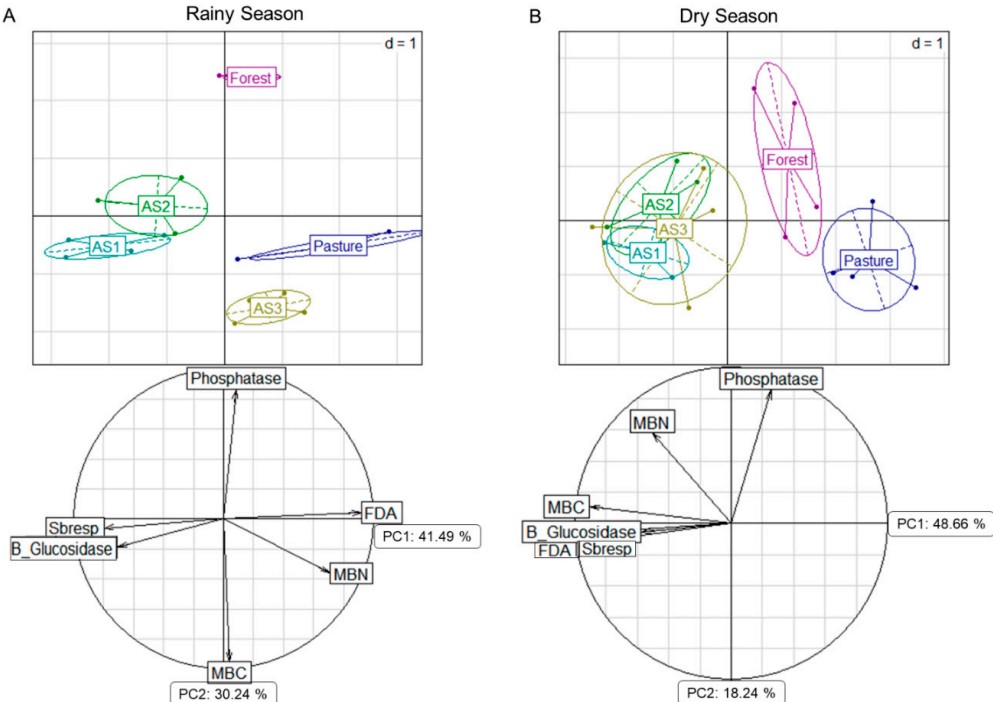

**Figure 3.** Between-class analysis of the five different land use using soil microbiological properties in (**A**) rainy season and (**B**) dry season (*p* = 0.001 by Monte Carlo permutation test). Variable-correlation circle of soil chemical and physical properties in (**A**) rainy season and (**B**) dry season. See Table 3 for additional description of abbreviations.



**Table 3.** Mean values for soil microbiological properties sampled on an experimental farm in the county of Sapucaia, Rio de Janeiro, Brazil at two time points, in the rainy season (April) and in the dry season (September) of 2018.

| Microbiological Variables | Rainy Season (April) | | | | | | Dry Season (September) | | | | | |
|---|---|---|---|---|---|---|---|---|---|---|---|---|
| | Forest | Pasture | AS1 | AS2 | AS3 | *p*-Value | Forest | Pasture | AS1 | AS2 | AS3 | *p*-Value |
| MBC (mg microbial C $kg^{-1}$ soil) | 339 [d] | 507 [b] | 530 [ab] | 429 [c] | 571 [a] | <0.001 | 667 [b] | 580 [c] | 708 [a] | 716 [a] | 727 [a] | <0.001 |
| | *22.5* | *20.6* | *26.6* | *27.4* | *22.1* | | *20.2* | *13.5* | *14* | *27.9* | *6.9* | |
| MBN (mg microbial N $kg^{-1}$ soil) | 42.9 [b] | 51.7 [ab] | 29.6 [c] | 28.9 [c] | 63.2 [a] | <0.001 | 70.6 [a] | 66.6 [a] | 72 [a] | 87.3 [a] | 75.1 [a] | |
| | *4.1* | *5.9* | *4.3* | *6.1* | *6.9* | | *15.6* | *7.8* | *5.2* | *6.3* | *15.2* | |
| Sbresp ($\mu$g C-$CO_2$ $g^{-1}$ $h^{-1}$) | 1.9 [a] | 1.7 [a] | 2.3 [a] | 2.1 [a] | 1.9 [a] | | 2.2 [bc] | 1.9 [c] | 3.3 [a] | 2.8 [ab] | 2.7 [abc] | <0.001 |
| | *0.2* | *0.5* | *0.4* | *0.2* | *0.4* | | *0.2* | *0.5* | *0.3* | *0.1* | *0.4* | |
| FDA ($\mu$g fluorescein $g^{-1}$ soil $h^{-1}$) | 114 [b] | 128 [a] | 96.2 [c] | 102.4 [bc] | 113 [b] | <0.001 | 101.2 [b] | 88.2 [b] | 130 [a] | 126 [a] | 126 [a] | <0.001 |
| | *5.2* | *3.22* | *2.7* | *5.4* | *9.8* | | *9.1* | *8.2* | *11.9* | *11.6* | *7.7* | |
| $\beta$-glucosidase ($\mu$mol $g^{-1}$ $h^{-1}$ p-nitrophenyl) | 6.5 [ab] | 5.7 [b] | 8.1 [a] | 7.1 [ab] | 7.1 [ab] | 0.072 | 9.9 [a] | 8.5 [a] | 10.5 [a] | 9.9 [a] | 10.2 [a] | |
| | *1.02* | *0.8* | *1.3* | *1.1* | *1.02* | | *1.02* | *0.8* | *1.3* | *1.1* | *1.02* | |
| Acidic Phosphatase ($\mu$mol $g^{-1}$ $h^{-1}$ p-nitrophenyl) | 8.6 [a] | 5.2 [b] | 5.6 [b] | 4.9 [b] | 5 [b] | <0.001 | 13.9 [a] | 5.7 [b] | 6.1 [b] | 6.3 [b] | 6.7 [b] | 0.008 |
| | *0.3* | *0.3* | *0.7* | *0.5* | *0.4* | | *4.5* | *2.4* | *2.9* | *2.4* | *2.8* | |

Values in italics below each mean represent the standard error from four measurements in each plot. Means with different letters have significantly different values according to Tukey tests.
Abbreviations: MBC, microbial biomass carbon; MBN, microbial biomass nitrogen; Sbresp, Soil Basal respiration; FDA, fluorescein diacetate hydrolysis.

### 3.3. Litter and Soil Dwelling Arthropods and Diversity Indices

A total of 7306 individuals were collected in the pitfall traps in the rainy season and another 6901 individuals in the dry season. These were separated into seven taxonomic groups, mainly at order and class level. In the rainy season, the most abundant organisms were Formicidae (ants, 39.9%), Collembola (19.8%), and Diptera (true flies, 14.7%). Ground-dwelling arthropods varied considerably across land use plots (Table 4). The highest total abundance was found under the pasture (42.1 ind. trap$^{-1}$ day$^{-1}$) and the lowest in AS2 (11.8 ind. trap$^{-1}$ day$^{-1}$). Taxonomic richness ranged from 6.25 to 6.88 on average per sampling point basis in each land use type; although there was no statistical difference, the richness tended to be lower in pasture and higher in agroforestry systems (AS1, AS3, AS2) and forest. Species diversity based on the Shannon Index (H) ranged from 1.20 to 1.99 and was highest in the agroforestry systems and forest and lowest in pasture. In the dry season, the most abundant organisms were Collembola (29.6%), Formicidae (ants, 23.3%), Diptera (true flies, 13.5%), Acari (8.2%), Araneae (spiders, 7.4%), Coleoptera (5.9%) across all land uses. The highest total activity was again found under pasture (31.6 ind. trap$^{-1}$ day$^{-1}$) and the lowest in the forest (15.3 ind. trap$^{-1}$ day$^{-1}$). Taxonomic richness ranged from 6.43 to 7 on average per sampling point basis in each land use type. Although there was no statistical difference, richness tended to be lower in pasture compared to forest and agroforestry systems. Species diversity based on the Shannon Index (H) ranged from 1.40 to 2.04 and was highest in the agroforestry systems and forest and lowest in pasture (Table 4). The groups that accounted for less than <5% of total abundance were included in Others at both times.

The results of the NMDS and PERMANOVA analyses showed that agroforestry plots and forest were separated from pasture. In April, the separation was largely associated with differences in Formicidae, Collembola, Araneae (Figure 4A). In September, the analysis again showed a clear separation between the land uses, with pasture clearly separated from forest and agroforestry systems. This separation was also related to Formicidae, Araneae as well as that of Coleoptera (Figure 4B).

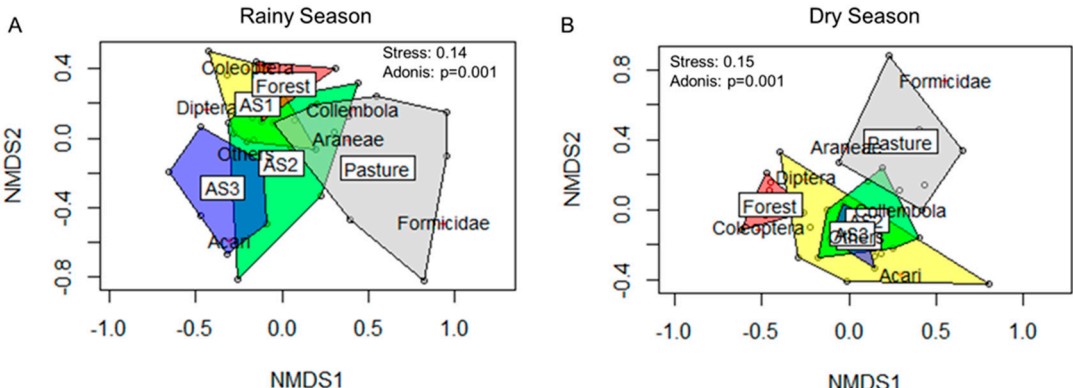

**Figure 4.** Nonmetric multidimensional scaling (NMDS) relating the soil fauna groups that representing more than 5% of total abundance, from plots sampled in rainy season (**A**) and dry season (**B**), respectively. Forest (red), Pasture (grey), AS1 (yellow), AS2 (green), AS3 (blue).

**Table 4.** Number of individuals per trap per day of the epigeal fauna communities, abundance and diversity on average per sampling point basis in each land use type on an experimental farm in the county of Sapucaia, Rio de Janeiro, Brazil at two time points, in the rainy season (April) and in the dry season (September) of 2018.

| Epigeal Fauna (Ind. trap⁻¹ day⁻¹) | Rainy Season (April) | | | | | | Dry Season (September) | | | | | |
|---|---|---|---|---|---|---|---|---|---|---|---|---|
| | Forest | Pasture | AS1 | AS2 | AS3 | *p*-Value | Forest | Pasture | AS1 | AS2 | AS3 | *p*-Value |
| Acari | 0.27 b | 0.53 b | 0.76 b | 0.94 b | 1.96 a | <0.001 | 0.65 c | 0.71 bc | 2.38 ab | 1.89 abc | 2.68 a | <0.01 |
| | 0.2 | 0.6 | 0.6 | 0.7 | 0.8 | | 0.3 | 0.6 | 1.9 | 0.7 | 1.4 | |
| Araneae | 1.11 a | 1.68 a | 1.09 a | 0.44 a | 0.63 a | | 1.07 a | 2.22 a | 1.75 a | 0.68 a | 1.86 a | |
| | 0.7 | 2.2 | 0.5 | 0.4 | 0.6 | | 0.5 | 3.5 | 1.4 | 0.5 | 0.8 | |
| Coleoptera | 1.22 a | 0.39 b | 1.06 ab | 0.65 ab | 0.57 ab | 0.013 | 2.37 a | 0.33 c | 1.36 b | 0.81 bc | 0.98 bc | <0.001 |
| | 1.2 | 0.4 | 0.6 | 0.4 | 0.5 | | 0.2 | 0.3 | 0.7 | 0.6 | 0.5 | |
| Collembola | 4.02 ab | 6.46 a | 3.36 ab | 4.46 ab | 2.25 b | 0.051 | 4.03 a | 7.6 a | 5.57 a | 7.19 a | 5.65 a | |
| | 1.2 | 3.6 | 2.8 | 3.5 | 1.3 | | 1.5 | 3.9 | 3.6 | 1.9 | 2.2 | |
| Diptera | 2.83 ab | 1.17 b | 4.14 ab | 1.42 b | 5.76 a | 0.016 | 4.15 a | 1.6 b | 2.42 ab | 2.81 ab | 2.51 ab | 0.073 |
| | 1.9 | 0.9 | 2.4 | 1.3 | 5.3 | | 1.7 | 1.1 | 2.2 | 1.9 | 0.7 | |
| Formicidae | 1.1 b | 30.3 a | 2.40 b | 1.89 b | 4.9 b | 0.002 | 1.21 b | 17.8 a | 2.39 b | 1.92 b | 1.44 b | <0.001 |
| | 0.4 | 32 | 1.8 | 1.5 | 7.9 | | 1 | 17.2 | 1.2 | 0.8 | 0.9 | |
| Others | 1.98 b | 1.56 b | 2.82 ab | 2.01 b | 4.92 a | 0.006 | 1.82 a | 1.38 a | 3.74 a | 2.64 a | 2.44 a | |
| | 0.42 | 1.01 | 2.20 | 1.04 | 2.95 | | 1.19 | 0.68 | 2.67 | 1.22 | 1.32 | |
| Total Abundance | 12.5 b | 42.1 a | 15.6 b | 11.8 b | 21 ab | 0.005 | 15.3 b | 31.6 a | 19.6 ab | 17.9 ab | 17.6 ab | 0.048 |
| | 4.1 | 32.3 | 6.8 | 6.2 | 15.2 | | 3.9 | 21.4 | 9.01 | 4.8 | 3.7 | |
| Richness (S) | 6.86 a | 6.25 a | 6.88 a | 6.62 a | 6.88 a | | 6.88 a | 6.43 a | 6.62 a | 6.75 a | 7.00 a | |
| | 0.38 | 1.16 | 0.35 | 0.52 | 0.35 | | 0.35 | 0.79 | 1.06 | 0.46 | 0.00 | |
| Shannon (H) | 1.99 a | 1.2 b | 1.93 a | 1.83 a | 1.96 a | <0.001 | 1.9 a | 1.4 b | 1.99 a | 1.9 a | 2.04 a | <0.001 |
| | 0.1 | 0.5 | 0.2 | 0.2 | 0.3 | | 0.2 | 0.3 | 0.4 | 0.2 | 0.1 | |

Values in italics below each mean represent the standard error from four measurements in each plot. Means with different letters have significantly different values according to Tukey tests.

### 3.4. Litter Chemical Properties

Litter biomass and chemistry from forest and agroforestry systems differed significantly in April, such that litter nutrient content (P, $Ca^{2+}$, $Mg^{2+}$ and $K^+$) of agroforestry systems was higher in relation to forest, while the forest presented the highest C:N ratio. In September there was a significant difference between treatment litter only for N content and C:N ratio. Litter biomass was highest in AS3 and forest in April, while no treatment differences where apparent in September (Table 5).

**Table 5.** Mean values for litter chemical properties sampled on an experimental farm in the county of Sapucaia, Rio de Janeiro, Brazil at two time points, in the rainy season (April) and in the dry seasons (September) of 2018. Samples were collected from five land uses: secondary forest, degraded pasture, and three agroforestry systems (AS1, AS2 and AS3).

| Litter Variables | Rainy Season (April) | | | | | Dry Season (September) | | | | |
|---|---|---|---|---|---|---|---|---|---|---|
| | Forest | AS1 | AS2 | AS3 | *p*-Value | Forest | AS1 | AS2 | AS3 | *p*-Value |
| C:N | 28.4 [a] | 25.4 [ab] | 21.5 [b] | 22.9 [b] | 0.026 | 23.4 [a] | 18.6 [b] | 18.3 [b] | 19.1 [b] | 0.028 |
| | *1.6* | *3.9* | *4.7* | *5.9* | | *1.8* | *1.4* | *0.7* | *1.2* | |
| P (mg kg$^{-1}$) | 0.05 [b] | 0.11 [a] | 0.11 [a] | 0.09 [a] | <0.001 | 0.03 [a] | 0.06 [a] | 0.06 [a] | 0.06 [a] | |
| | *0.02* | *0.02* | *0.02* | *0.02* | | *0.01* | *0.02* | *0.01* | *0.01* | |
| $Ca^{2+}$ (mg kg$^{-1}$) | 94.65 [b] | 156.7 [a] | 148.7 [ab] | 136.9 [a] | 0.009 | 99.3 [a] | 129 [a] | 101 [a] | 101 [a] | |
| | *9.7* | *5.2* | *15.4* | *15.6* | | *29.1* | *9.7* | *12.7* | *12.4* | |
| $Mg^{2+}$ (mg kg$^{-1}$) | 18.2 [b] | 26.2 [ab] | 23.2 [ab] | 41.3 [a] | 0.017 | 23.2 [a] | 28.1 [a] | 20.0 [a] | 25.5 [a] | |
| | *3.3* | *8.9* | *2.5* | *7.5* | | *7.8* | *2.9* | *7.3* | *7.2* | |
| $K^+$ (mg kg$^{-1}$) | 9.99 [b] | 16.4 [a] | 10.4 [b] | 15.3 [ab] | 0.008 | 17.2 [a] | 29.9 [a] | 20.3 [a] | 39.3 [a] | |
| | *0.7* | *3.1* | *1.9* | *4.5* | | *7.7* | *3.6* | *4.2* | *27.6* | |
| Biomass (kg ha$^{-1}$) | 1578 [a] | 1366 [b] | 1112 [c] | 1499 [ab] | <0.001 | 1410 [a] | 1086 [a] | 1261 [a] | 1193 [a] | |
| | *64.8* | *60.6* | *56.3* | *79.1* | | *64* | *149* | *208* | *264* | |

Values in italics below each mean represent the standard error from four measurements in each plot. Means with different letters have significantly different values according to Tukey tests.

### 3.5. Relationships Between Litter Quality, SOM and Key Soil Quality Parameters

When accounting for the different sampling times, litter quality and SOM were significantly related to a number of soil biological, chemical and physical parameters (Table 6). For example, litter C:N ratio was negatively correlated with invertebrate abundance, MBC, available K and P, pH, and BD, and was positively correlated with phosphatase activity and aggregate stability (MWD). The effect of litter C:N ratio on the soil microbial parameters FDA and Sbresp appeared to depend on sampling time (interaction $p \leq 0.010$), such that FDA increased (and Sbresp did not change) with increasing C:N ratio for samples collected in the rainy season (April) and both parameters decreased with C:N in the dry season (September). SOM was positively related to MBC, Sbresp, β-glucosidase activity, available K, pH and CEC, and negatively correlated with BD. There were also marginally significant ($p < 0.07$) positive relationships between SOM and macrofauna richness, Shannon diversity, and available P. Significant interactions between sampling time and SOM content ($p < 0.05$), suggest that the relationship with FDA and pH depend on the sampling time in question (Table 6). Sampling time was an important predictor of many of the response variables considered, both when looking at relationships with litter quality and SOM content (Table 6).

**Table 6.** Model results depicting the relationships between litter quality (C:N ratio) and soil organic matter (SOM) with soil quality variables from soils collected from five land uses and two sampling times (Rainy or dry season) on an experimental farm in the county of Sapucaia, Rio de Janeiro, Brazil.

| Soil Response Variable | C:N Ratio | C:N Effect Direction * | Sampling Time | C:N × Time | SOM | SOM Effect Direction * | Sampling Time | SOM × Time |
|---|---|---|---|---|---|---|---|---|
| Abundance | 0.029 | - | ns | ns | ns | | ns | ns |
| Richness | ns | | ns | ns | 0.061 | + | ns | ns |
| Shannon | ns | | ns | ns | 0.058 | + | ns | ns |
| MBC | 0.025 | - | <0.001 | ns | 0.009 | + | <0.001 | ns |
| MBN | ns | | <0.001 | ns | ns | | <0.001 | ns |
| FDA | 0.030 | Apr+, Sep− | ns | <0.001 | ns | Apr−, Sep+ | ns | 0.002 |
| Sbresp | 0.003 | Apr 0, Sep− | ns | 0.010 | 0.034 | + | <0.001 | ns |
| β-glucosidase | ns | | <0.001 | ns | <0.001 | + | <0.001 | ns |
| Phosphatase | 0.003 | + | <0.001 | ns | 0.093 | - | ns | ns |
| K | 0.002 | - | 0.001 | ns | <0.001 | + | ns | ns |
| CEC | ns | | 0.017 | 0.010 | 0.035 | + | 0.009 | ns |
| pH | 0.045 | - | 0.009 | ns | 0.002 | Apr+, Sep 0 | ns | 0.021 |
| P | 0.021 | - | <0.001 | ns | 0.057 | + | 0.023 | ns |
| BD | 0.030 | - | ns | ns | 0.012 | - | ns | ns |
| MWD | 0.022 | + | 0.088 | ns | ns | | ns | ns |
| POXC | ns | | ns | ns | ns | | ns | ns |
| SOM | ns | | 0.081 | ns | NA | | NA | NA |

* refers to the direction of the correlation overall or for the different sampling times (in case of significant interaction); Positive relationship (+) or negative relationship (−).

## 4. Discussion

### 4.1. Soil Chemical and Physical Properties Across Different Land Uses

Clear differences in soil chemical properties were evident between the forest, pasture and agroforestry systems plots. Soils under forest and pasture generally had lower fertility (in terms of pH, SOC, POXC, available K) than the agroforestry system plots (Table 2). While not always significant for the univariate tests, multivariate comparisons showed a high degree of separation between agroforestry systems and the pasture and forestry plots (Figure 2A,B). The agroforestry systems also tended to have higher levels of available P and CEC than pasture, thus contributing to the overall higher soil fertility in the agroforestry plots. We note that the agroforestry systems received an initial addition of rock phosphate to the banana planting pits during the first year of establishment. This may have contributed to the higher P availability and pH observed in those systems [57], but these initial P inputs may have also contributed to SOM by increasing overall system productivity and the return of organic residues to the soil, thus further contributing to soil fertility impacts. Another factor contributing to the higher levels of soil fertility under agroforestry system is the high diversity of tree species and improved litter quality derived from these trees in the agroforestry system plots. For example, *Tithonia diversifolia*, which was included in all of the treatments, is known to accumulate high concentrations of N, P and K in its leaves [58]. Residue from such plants can contribute to soil fertility status through nutrient mobilization and return as well as by contributing to SOM, which can increase the availability of P in acid soils by blocking the P adsorption sites on mineral surfaces [59]. Additionally, the greater presence of legumes under the agroforestry systems likely contributed to litter quality, as is evidenced by C:N ratios of litter from forest vs. agroforestry systems. Residues that are rich in N are thought to enhance the C use efficiency of decomposer organisms [60,61], resulting in a higher microbial biomass [60] and eventual stabilization of SOC. Our findings thus lend support to the potential role of agroforestry systems for sequestering C and supporting overall soil fertility in agricultural lands [24,25].

When considering soil physical properties, the forest had the highest aggregate stability and among the lowest bulk density values, suggesting that these soils may support higher water infiltration rates and improved erosion control [62,63]. Meanwhile, the pasture had the highest bulk density, indicating potential compaction issues (Table 2). This is likely associated with poor grazing and soil fertility management, as it had received no fertilization and had been under continuous grazing for many years. The lack of nutrient inputs and other management interventions to maintain both above- and belowground productivity has been shown to negatively affect soil structure and overall fertility in other tropical pasture systems [64]. This degraded condition is likely representative of many Brazilian pastures, since 80% have been considered to be in some state of degradation [4]. At the same time, we note that the agroforestry system plots all had bulk densities that were more similar to the forest soil, suggesting potential amelioration or avoidance of compaction issues since establishment of the agroforestry systems on degraded pastures 8 years prior. However, as stated above, due to the lack of true replication in this study, we cannot draw firm conclusions about how management impacts soil properties. Nevertheless, we note that prior to the establishment of the agroforestry systems, these plots were under a nearly identical management regime as the adjacent pasture plot. Additionally, given that there are no significant differences in soil texture between plots, it is likely that the soils began in a similar state prior to agroforestry system implementation and thus likely that many of the differences observed between plots are due at least in part to management. While important insights can be gained from this study, our findings and any causal inferences suggested here should be interpreted with caution.

### 4.2. Soil Biological Properties

Beyond differences in soil physiochemical properties, the management plots evaluated here indicated clear differences in soil biological properties. For example, the agroforestry systems supported generally higher levels of microbial biomass C and soil respiration (Table 3), especially in

the dry season (September), several weeks after pruning. A similar trend was also apparent for FDA and β-glucosidase activity, suggesting that the high inputs of relatively high-quality residues in agroforestry systems encouraged microbial growth and activity. Soils under secondary forest generally had intermediate values between agroforestry systems treatments and pasture for many microbial parameters. While the forest, comprised largely of semideciduous species, also deposited high amounts of residues as senescent litter in the dry season (as indicated by high standing litter biomass), this material was of much lower quality than the agroforestry systems residues from pruning (Table 5). This senesced litter is not likely to decompose as rapidly and stimulate microbial activity to the same extent as green leaves [65]. At the same time, the forest soil showed the highest values of acid phosphatase at both sampling times and this was a key factor differentiating the forest from the other management plots (Figure 3A,B). We suspect that this is associated with the generally lower levels of available P in the forest system and the high inputs of low quality litter, since both plants and microbes under such conditions are likely to respond by producing phosphatase to stimulate P mineralization [66]. Among the agroforestry systems, AS3 displayed the highest microbial biomass (Table 3). This may be related to the higher density of the biomass species *T. diversifolia* in this system. Jama et al. [58] working in tropical Africa, found this species to support increased microbial biomass and soil biological activity. Overall, we note that the three agroforestry systems supported relatively similar microbial properties relative to the forest or pasture plots considered here.

The assessment of ground dwelling arthropods demonstrated clear differences between land use systems, especially between the forest and pasture plots (Figure 4A,B). In general, there was higher diversity in plots with trees than was observed in the pasture, while the pasture had the highest abundance of arthropods, comprised mostly of ants (Table 4). We suggest that this may related with diversity of food resources. When there is a reduction in the diversity of food resources, some ground-dwelling arthropod groups can establish themselves quickly and dominate the community, as is often observed with social insects, such as ants [67]. The complexity of the litter structure is often a good predictor of the abundance and diversity of soil litter fauna. In a survey of soil quality parameters, including ground-dwelling arthropods in Nicaragua, Rousseau et al. [21] found forests to have higher diversity than nearby pasture systems, while a maize-bean agroforestry system displayed intermediate levels of diversity. This same study found that another social insect, termites, were significantly higher in the pasture than in secondary forest. They related these findings to increased habitat complexity and organic matter inputs in the tree-based systems [21]. Soils with high litter biomass and diversity (from multiple tree species) are expected to have a higher diversity and abundance of soil fauna groups since they allow for higher numbers of microhabitats and therefore increase niche differentiation between groups [68]. We suspect that the high diversity of ground-dwelling invertebrates observed in the agroforestry systems is related to the similarity of these systems, in terms of vegetative structure and the presence of a developed litter layer. Beyond effects on overall diversity and the dominant groups, we noted clear differences in key functional groups of macrofauna. For example, Coleoptera (beetles) tended to be more abundant in the forest than in the other systems, and it may be that this group is more sensitive to disturbance and thus a good indicator of forest restoration [69].

### 4.3. Linkages Between Litter Quality, SOM and Soil Quality Parameters

We found evidence for linkages between litter quality, SOM and several important soil quality metrics, highlighting the importance of litter input type and processing to ecological functioning. Specifically, we found the C:N ratio to be negatively correlated with microbial biomass C and the abundance of ground dwelling arthropods (Table 6), both important indicators of soil biological activity. These negative correlations suggest that high quality litter inputs (i.e., low C:N ratio) are important for stimulating decomposition [70] and mineralization processes [60] and the activity of a range of soil organisms [71,72] that depend the resulting increases in nutrient availability. Changes to nutrient availability, and implications for overall system productivity, are further evidenced by increased availability of P and K with decreasing C:N. Surprisingly, we found that aggregate stability (MWD)

tended to increase with C:N ratio. This was somewhat surprising since aggregation is often associated with increased soil biological activity [73]. It may be that higher C:N litter stimulated increased growth of fungal hyphae in soils, which can be important for the formation of macroaggregates [74,75].

Similar to the effects of litter quality, SOM was related to soil biological activity and multiple soil properties. SOM was positively related to microbial biomass C, respiration, and β-glucosidase activity and also showed marginally significant relationships with the diversity of ground dwelling arthropods. This finding supports the general notion that, together with litter inputs, SOM is an essential resource base for not only decomposer organisms, but entire soil food webs [76,77], and is thus a key driver of multiple soil biological processes. We also note that the observed positive correlations between SOM and available K and P, as well as CEC, point to the importance of SOM in both water and nutrient cycling that has been reported elsewhere [78,79].

Given the marked impacts of SOM and litter quality on soil functioning suggested by our findings, both represent key management objectives in the development of agroforestry systems to restore degraded lands. Litter quality, in particular, can be readily managed via tree species selection and might play an important role in facilitating soil restoration through effects on biological communities and the functions they regulate. The use of leguminous tree species for biomass production in these systems is particularly relevant since many of them are associated with $N_2$-fixing bacteria, suppling N to the system and enhancing nutrient cycling and soil fertility through high-quality litter that supports soil biological communities. Additionally, pruning management is another practice that regulates the quality and amount of litter inputs entering the system and strategic timing is needed in order to balance litter inputs and nutrient cycling with productivity.

## 5. Conclusions

The physicochemical and biological assessment of these land uses reveal that soil under agroforestry systems generally had higher soil fertility status, as well as SOM and biological activity than either pasture or forest. The linkages between litter quality, SOM and soil parameters suggest that high quality litter inputs (i.e., low C:N ratio) together with SOM are important for stimulating biological activity and multiple soil properties with future implications for soil restoration. While causal inferences of management cannot be drawn from this study, our findings lend support to the idea that establishment of complex agroforestry systems in Brazil is likely to support soil quality and restoration goals. This further supports environmental legislation suggesting agroforestry systems as a viable option for Brazil to restore degraded lands and comply with international commitments to reduce greenhouse gas emissions, move towards low-C agriculture, and consequently contribute to improving global food security. Future studies are needed to confirm causal linkages between agroforestry management and soil quality indicators as well as to understand contributions to above and belowground C stocks, and to better quantify the potential of the agroforestry systems to restore soils across different levels of degradation.

**Author Contributions:** Conceptualization and methodology, P.S.M., E.Z. and M.G.P.; data collection and laboratorial analysis, P.S.M., M.A.F., G.M.C., S.S.L. and F.C.B.; statistical analysis, P.S.M., C.K., J.M.D. and S.J.F.; resources, E.Z., M.G.P., F.C.B.; writing, P.S.M. and S.J.F.; review and commentary, S.J.F., M.G.P. and J.M.D.; visualization, C.K.; supervision, E.Z. and M.G.P. All authors have read and agreed to the published version of the manuscript.

**Funding:** This research received no external funding.

**Acknowledgments:** This study was supported by a scholarship to the first author from the Brazilian Higher Education Personnel Improvement Coordination (CAPES).

**Conflicts of Interest:** The authors declare no conflict of interest.

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
