# Peer review of "Linkages among Soil Properties and Litter Quality in Agroforestry Systems of Southeastern Brazil"

_sustainability, doi:10.3390/su12229752_

Round 1

Reviewer 1 Report

The aim of this research were to: 1) assess soil chemical, physical and biological properties across distinct land uses (degraded pasture, agroforestry, and secondary forest); and 2) understand relationships between litter quality, soil  organic matter (SOM), and key soil quality parameters in the Brazilian Atlantic Rainforest. Authors findings suggest that agroforestry systems can help support soil  biological, chemical and physical properties and that the litter quality may be an important driver  of their effects and potential contributions to soil restoration in the region.

In my opinion this manuscript is very interesting but has some problem that need correcting. My specific comments are as follow:

  1. In subsection 3.2 the Authors used the MBC and MBN abbreviations (line 292, table 3 etc ....) to discuss the results. These abbreviations are nowhere described. If they are used for the first time in the text, they should be explained. In my opinion that they should already be marked in the section materials and methods. The same applies to the abbreviation "Sbresp" !! The subsection on materials and methods should be improved

  1. In the discussion section, The authors try to explain the high level of acid phosphatase activity in forests. They only indicate an increase in its secretion by microorganisms, which is a mistake (line 451). It is necessary to remember that phosphorus-deficient plants are characterised by increased secretion of acid phosphatase through the root system into the soil more than microorganisms .

Reviewer 2 Report

Dear Authors, 

The work is interesting; it is very well structured and the research is relevant. The only drawback that I find is that you have not made replications. However, as you recognize, I understand that this article is the first step in a longer-term investigation, so I consider the preliminary presentation of these results adequate, since they are of interest.

Kind regards,

The Reviewer

Reviewer 3 Report

I strongly recommend the article.

The article is well organised, the reader is going smoothly to te main objectives of the pproposed study. The objectives of this research were well introduces from the beginning and the reader was aware about the number of assesd soils and the set of the chemical, physical and biological properties.

The manner of data treatment to go more insighn and for better understanding of the  relationships between  soil organic matter (SOM), and key soil quality parameters in the Brazilian Atlantic Rainforest by the propose metheod is a good approach for looking for the hidden pattrens within the data.

What I will recoment is to include and the Cluster analys as well with the K-mean like a techniques for the next study.
